

# Unified methods in collecting, preserving, and archiving coral bleaching and restoration specimens to increase sample utility and interdisciplinary collaboration

Rebecca Vega Thurber[1], Emily R. Schmeltzer[1], Andréa G. Grottoli[2], Robert van Woesik[3], Robert J. Toonen[4], Mark Warner[5], Kerri L. Dobson[2], Rowan H. McLachlan[1,2], Katie Barott[6], Daniel J. Barshis[7], Justin Baumann[8], Leila Chapron[2], David J. Combosch[9], Adrienne MS Correa[10], Thomas M. DeCarlo[11], Mary Hagedorn[4,12], Laetitia Hédouin[13], Kenneth Hoadley[14], Thomas Felis[15], Christine Ferrier-Pagès[16], Carly Kenkel[17], Ilsa B. Kuffner[18], Jennifer Matthews[19], Mónica Medina[20], Christopher Meyer[21], Corinna Oster[15], James Price[2], Hollie M. Putnam[22] and Yvonne Sawall[23]

[1] Department of Microbiology, Oregon State University, Corvallis, OR, United States
[2] School of Earth Sciences, Ohio State University, Columbus, OH, United States
[3] Institute for Global Ecology, Florida Institute of Technology, Melbourne, Fl, United States
[4] Hawai'i Institute of Marine Biology, University of Hawai'i at Mānoa, Kāne'ohe, HI, United States
[5] School of Marine Science and Policy, University of Delaware, Lewes, DE, United States
[6] Department of Biology, University of Pennsylvania, Philadelphia, PA, United States
[7] Department of Biological Sciences, Old Dominion University, Norfolk, VA, United States
[8] Biology Department, Bowdoin College, Brunswick, ME, United States
[9] Marine Laboratory, University of Guam, Mangilao, Guam
[10] BioSciences Department, Rice University, Houstan, TX, United States
[11] College of Natural and Computational Sciences, Hawai'i Pacific University, Honolulu, HI, United States
[12] Conservation Biology Institute, Smithsonian, Kāne'ohe, HI, United States
[13] Centre de Recherches Insulaires et Observatoire de l'Environnement, Chargée de Recherches CNRS, Papetō'ai, Moorea, French Polynesia
[14] Department of Biological Sciences, University of Alabama – Tuscaloosa, Tuscaloosa, AL, United States
[15] MARUM – Center for Marine Environmental Sciences, University of Bremen, Bremen, Germany
[16] Marine Biology Department, Coral Ecophysiology team, Centre Scientifique de Monaco, Monaco
[17] Department of Biological Sciences, University of Southern California, Los Angeles, CA, United States
[18] U.S. Geological Survey, St. Petersburg, FL, United States
[19] Climate Change Cluster, University of Technology Sydney, Sydney, Australia
[20] Department of Biology, Pennsylvania State University, University Park, PA, United States
[21] Department of Invertebrate Zoology, National Museum of Natural History, Smithsonian, Washington DC, United States
[22] Department of Biological Sciences, University of Rhode Island, Kingston, RI, United States
[23] Bermuda Institute of Ocean Sciences, St. George's, St. George's, Bermuda

Corresponding author
Rebecca Vega Thurber,
rebecca.vega-thurber@oregonstate.edu

## ABSTRACT

Coral reefs are declining worldwide primarily because of bleaching and subsequent mortality resulting from thermal stress. Currently, extensive efforts to engage in more holistic research and restoration endeavors have considerably expanded the

techniques applied to examine coral samples. Despite such advances, coral bleaching and restoration studies are often conducted within a specific disciplinary focus, where specimens are collected, preserved, and archived in ways that are not always conducive to further downstream analyses by specialists in other disciplines. This approach may prevent the full utilization of unexpended specimens, leading to siloed research, duplicative efforts, unnecessary loss of additional corals to research endeavors, and overall increased costs. A recent US National Science Foundation-sponsored workshop set out to consolidate our collective knowledge across the disciplines of Omics, Physiology, and Microscopy and Imaging regarding the methods used for coral sample collection, preservation, and archiving. Here, we highlight knowledge gaps and propose some simple steps for collecting, preserving, and archiving coral-bleaching specimens that can increase the impact of individual coral bleaching and restoration studies, as well as foster additional analyses and future discoveries through collaboration. Rapid freezing of samples in liquid nitrogen or placing at −80 °C to −20 °C is optimal for most Omics and Physiology studies with a few exceptions; however, freezing samples removes the potential for many Microscopy and Imaging-based analyses due to the alteration of tissue integrity during freezing. For Microscopy and Imaging, samples are best stored in aldehydes. The use of sterile gloves and receptacles during collection supports the downstream analysis of host-associated bacterial and viral communities which are particularly germane to disease and restoration efforts. Across all disciplines, the use of aseptic techniques during collection, preservation, and archiving maximizes the research potential of coral specimens and allows for the greatest number of possible downstream analyses.

**Subjects** Biodiversity, Conservation Biology, Ecology, Marine Biology, Biological Oceanography
**Keywords** Coral, Reef, Provenance, Storage, Methodology, Protocols, Pipelines, Analytics, Physiology, Omics

## INTRODUCTION

Coral reefs provide sustenance, goods, and services for coastal communities worldwide and likely harbor more than one third of all marine species (*Fisher et al., 2015*). However, corals and reef frameworks are increasingly being degraded due to anthropogenic disturbances (*Intergovernmental Panel on Climate Change, 2022*). Climate change has severely affected coral reef health on a global scale, primarily through increased sea surface temperatures leading to devastating coral bleaching events. The increased frequency and intensity of these events reduces the capacity for reef recovery and restoration efforts (*Heron et al., 2016*; *van Hooidonk et al., 2016*; *Sully et al., 2019*), and successive bleaching events have decreased live coral cover by up to 60% in some localities (*Miller et al., 2009*; *Raymundo et al., 2019*; *Dalton et al., 2020*). Up to one third of all reef-building corals species may be at risk of extinction from the combined effects of bleaching and local stressors such as nutrient pollution, overfishing, and habitat destruction (*Pandolfi et al., 2003*; *Carpenter et al., 2008*; *Plaisance et al., 2011*; *Hughes et al., 2017*, *2018*, *2019*). Given the increased frequency and severity of bleaching events, scientists and restoration practitioners need to

study coral bleaching and disease more. One way to achieve greater efficiency is through the implementation of a common framework recently developed for coral bleaching experiments (*Grottoli et al., 2021*). Another is by reducing the number of duplicative efforts more broadly and maximizing the number of analyses that can be performed on sampled specimens through greater collaboration.

## Identifying common methodological pipelines in collecting, preserving, and archiving

Between 2014 and 2021 over 20,000 coral specimens and samples were collected for bleaching studies (*McLachlan et al., 2021*), many of which are suitable for additional analyses that could address new questions concerning various aspects of bleaching. The technology and methods commonly used in coral biology research have quickly progressed in recent decades (*Cziesielski, Schmidt-Roach & Aranda, 2019*; *Grottoli et al., 2021*). The combination of traditional and modern genomic insights, physiological metrics, and microscopy and imaging analytics have together given scientists an ever-expanding toolkit to interrogate the mechanisms and results of coral bleaching and restoration efforts at the subcellular, cellular, tissue, and organismal levels. Integration of these approaches thus allows individual specimens to be used for multiple downstream applications and expands the potential utility of every coral sample collected. Despite this, scientists and practitioners tend to sample, preserve, and archive specimens in a manner specific to their own specialized applications or aims, and on average only conduct one or two downstream analyses per study (*McLachlan et al., 2021*). However, it is unclear how many or how often archived samples are utilized. Yet, limits exist on how many tools individual researchers can manage, conduct, and financially support. Trained in increasingly complicated fields of study, it is impractical for any one scientist, or even a team of scientists, to have the breadth of knowledge, skills, and resources to conduct the full range of possible Omics, Physiology, and Microscopy and Imaging analyses on any given set of specimens. However, with effective documentation during sampling (*Grottoli et al., 2021*), coupled with strategic preserving and archiving decisions, specimens could be available to additional research teams, who could increase the number of analyses ultimately conducted on a given set of samples, contributing to a better understanding of bleaching mechanisms with less sampling and experimental damage to reefs.

As the numbers and expertise of scientific investigators expand, so do the tools, methods, and perspectives at their disposal. We brought together investigators from around the world to further synthesize research methods in order to identify low-cost and practical ways to share specimens, reduce duplicative efforts, and increase the end-use potential of samples generated in coral bleaching research and restoration programs (Fig. 1). We identified and consolidated working pipelines that could (1) expand the number of potential analyses on currently archived samples, and (2) assist in future project planning to maximize the number of potential downstream analyses while minimizing any extra work, time, or funds required. While no single methodological pipeline can be all-inclusive, several critical steps in these methodological pipelines were found to optimize

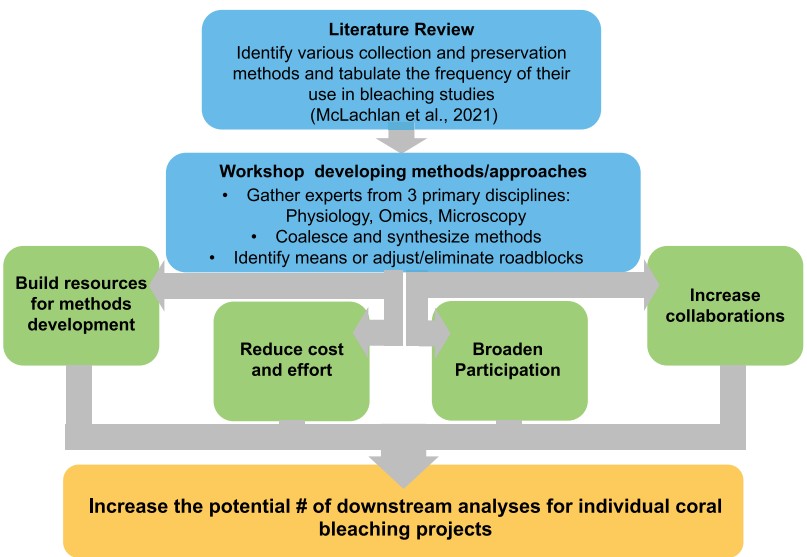

**Figure 1** Flow chart of conceptual design for workshop on methods of collecting preserving and archiving coral bleaching specimen.

the potential utility of each coral specimen within the constraints of a given study design (Fig. 2).

## Consolidating methods for broadening participation

Cheap and unifying methods can serve to increase participation and inclusion in coral bleaching and restoration research, particularly for those with minimal funding. Clear, simple guidelines for specimen and sample collection, manipulation, and preservation can also make it easier for experts working on parallel questions in non-coral systems to bring their hypotheses and approaches to bear on the coral bleaching and restoration fields. Adapting and expanding sampling, preserving, and archiving of specimens in ways that allow for additional downstream analyses can generate research opportunities for early career scientists and students, providing a mechanism for additional collaboration and more entry points into the field of coral research, as well as creating new opportunities for collaborations and networking between researchers with distinct yet complementary areas of inquiry, thereby fostering advances and new ideas within the field. These efforts support the inclusion of researchers in the field who may not currently conduct marine fieldwork due to lack of access to resources (*e.g.*, funding, SCUBA gear, boat access, laboratory equipment), training (*e.g.*, scientific dive certifications), and/or physical or logistical capability. A separate challenge in promoting diversity and inclusion in the broader field of coral research is to connect researchers that have samples with other scientists and managers (including undergraduate trainees and volunteers) from diverse disciplines and backgrounds that can run additional analyses. A database of samples and researchers (and their research interests/skill sets) could be useful in identifying and jump-starting fruitful collaborations and sample sharing. Numerous community-based resources can also provide data storage options to both facilitate data archiving and reuse, including those specific to coral research, restoration and biodiversity (*e.g.*, GEOME (*Deck et al., 2017*;

| A | B | C | D | E | F | G | H | I | J | K | L | M | N | O | P | Q | R | S | T | U | V | W | X | Y | Z | AA | AB | AC | AD | AE | AF | AG | AH | AI | AJ | AK | AL | AM | AN |
|---|---|---|---|---|---|---|---|---|---|---|---|---|---|---|---|---|---|---|---|---|---|---|---|---|---|---|---|---|---|---|---|---|---|---|---|---|---|---|---|
| 1 | **Collection & Archiving Steps** | | **Downstream analyses** | | | | | | | | | | | | | | | | | | | | | | | | | | | | | | | | | | | | |
| 2 | | | **Omics** | | | | | | | | **Physiology** | | | | | | | | | | **Microscopy & Imaging** | | | | | | | | | | | | | | | | | | |
| 3 | *After removing coral specimen from the reef, it is immediately placed into one of the following:* | *Specimen is then archived at this temperature (see key):* | DNA analyses | | RNA analyses | | Epigenetics methylation approaches | Epigenetics histone-chromatin approaches | Proteomics or enzyme assays | Metabolomics | Chlorophyll | HPLC pigment analyses | MAA | Total soluble lipid & lipid classes | Total soluble protein | Carbohydrates | Biomass ash-free dry weight | Non-isotopic elemental composition | Stable light isotopes | | SEM surface ultrastructure | | | | TEM internal ultrastructure | | | NanoSIMS internal ultrastructure | Histology | | Nucleic acid probes/FISH | Immunolocalization | CT scanning | X-ray imaging | Epifluorescence microscopy with dye | XRF scanning, micro XRF, EA | Raman spectroscopy | Hemacytometry, flow cytometry, automated cell counters | ↓ Percent *Optimal (O) or Acceptable (A)* |
| 4 | | | ***SPECIMEN TYPE (see key)*** | | | | | | | | | | | | | | | | | | | | | | | | | | | | | | | | | | | | |
| 5 | ↓ | ↓ | CH AS | M | CH AS | M | CH AS M | CH AS M | CH AS M | CH AS M | AS | AS | CH AS M | CH AS | CH AS | CH AS | CH AS | SK | SK | CH AS | CH AS | G | SK | M | CH AS | G | M | CH AS | CH AS | G | CH AS | CH AS | SK | SK | SK | SK | SK | AS | |
| 6 | Dry oven | RT | N | N | N | N | N | N | N | N | N | N | ? | N | N | N | N | O | O | ? | N | N | A | ? | N | N | N | N | N | N | N | N | A | A | A | O | A | N | 22% |
| 7 | Liquid nitrogen (-196℃) | F | O | O | O | O | O | A | O | O | O | O | O | O | O | O | A | A | A | A | N | N | A | N | N | N | N | U | N | N | U | N | A | A | A | A | U | U | 64% |
| 8 | | RF | U | N | N | N | ? | ? | N | U | ? | N | ? | N | N | N | N | A | A | N | N | N | A | N | N | N | N | U | N | N | U | N | A | A | A | A | ? | U | 19% |
| 9 | | RT | U | N | N | N | N | N | N | N | N | N | ? | N | N | N | N | A | A | N | N | N | A | N | N | N | N | N | N | N | N | N | A | A | A | A | ? | U | 19% |
| 10 | | D | N | N | N | N | N | N | N | N | N | N | ? | ? | N | ? | ? | A | A | ? | N | N | A | N | N | N | N | N | N | N | N | N | A | A | A | A | ? | N | 19% |
| 11 | Conventional freezer (-20 to -39℃) | F | A | A | U | U | A | A | A | A | A | A | U | U | A | A | A | A | A | A | N | N | A | N | N | N | N | U | N | N | U | N | A | A | A | A | ? | A | **56%** |
| 12 | | RF | U | N | N | N | N | N | N | U | ? | N | ? | N | N | N | N | A | A | N | N | N | A | N | N | N | N | U | N | N | U | N | A | A | A | A | ? | A | 22% |
| 13 | | RT | U | N | N | N | N | N | N | N | N | N | ? | N | N | N | N | A | A | N | N | N | A | N | N | N | N | N | N | N | N | N | A | A | A | A | ? | N | 19% |
| 14 | | D | N | N | N | N | N | N | N | N | N | N | ? | ? | N | ? | ? | A | A | ? | N | N | A | N | N | N | N | N | N | N | N | N | A | A | A | A | ? | N | 19% |
| 15 | Ultra-low freezer (-80 to -40 ℃) | F | A | A | U | U | O | O | A | O | A | O | U | O | O | O | A | A | A | A | N | N | A | N | N | N | N | U | N | N | U | N | A | A | A | A | ? | U | 56% |
| 16 | | RF | U | N | N | N | N | N | N | U | ? | N | ? | N | N | N | N | A | A | N | N | N | A | N | N | N | N | U | N | N | U | N | A | A | A | A | ? | U | 19% |
| 17 | | RT | U | N | N | N | N | N | N | N | N | N | ? | N | N | N | N | A | A | N | N | N | A | N | N | N | N | N | N | N | N | N | A | A | A | A | ? | U | 19% |
| 18 | | D | N | N | N | N | N | N | N | N | N | N | ? | ? | N | ? | ? | A | A | ? | N | N | A | N | N | N | N | N | N | N | N | N | A | A | A | A | ? | N | 19% |
| 19 | Ethanol | F | A | A | A | A | ? | ? | N | N | ? | N | ? | N | N | N | N | N | N | ? | N | N | A | N | N | N | N | N | N | N | N | N | A | A | ? | ? | ? | N | 19% |
| 20 | | RF | A | A | A | A | ? | ? | N | N | ? | N | ? | N | N | N | N | N | N | ? | A | A | A | ? | ? | A | ? | U | U | U | N | U | A | A | ? | ? | ? | N | 28% |
| 21 | | RT | A | A | A | A | N | N | N | N | ? | N | ? | N | N | N | N | N | N | ? | A | A | A | ? | N | A | ? | U | U | U | N | U | A | A | ? | ? | ? | N | 28% |
| 22 | | D | N | N | N | N | N | N | N | N | N | N | ? | N | N | N | N | N | N | ? | ? | ? | A | ? | N | ? | ? | N | N | N | N | N | A | A | ? | ? | ? | N | 8% |
| 23 | Methanol | F | ? | ? | ? | ? | ? | ? | N | A | ? | N | ? | N | N | N | N | N | N | ? | N | N | A | N | N | N | N | N | N | N | N | N | A | A | ? | ? | ? | N | 11% |
| 24 | | RF | ? | ? | ? | ? | ? | ? | N | U | ? | N | ? | N | N | N | N | N | N | ? | A | A | A | N | ? | A | ? | U | U | U | N | ? | A | A | ? | ? | ? | N | 17% |
| 25 | | RT | ? | ? | ? | ? | N | N | N | U | ? | N | ? | N | N | N | N | N | N | ? | A | A | A | N | N | A | ? | U | U | U | N | N | A | A | ? | ? | ? | N | 17% |
| 26 | | D | N | N | N | N | N | N | N | N | N | N | ? | N | N | N | N | N | N | ? | ? | ? | ? | N | N | ? | ? | N | N | N | N | N | A | A | ? | ? | ? | N | 6% |
| 27 | RNAlater, salt saturated DMSO | F | O | O | A | A | A | A | U | N | ? | N | ? | N | N | N | N | N | N | ? | N | N | A | N | N | N | N | N | N | N | N | N | A | A | ? | ? | ? | N | **25%** |
| 28 | | RF | A | A | A | A | N | N | N | N | ? | N | ? | N | N | N | N | N | N | ? | A | A | A | ? | ? | A | ? | N | N | N | N | N | A | A | ? | ? | U | N | **28%** |
| 29 | | RT | A | A | A | A | N | N | N | N | ? | N | ? | N | N | N | N | N | N | ? | A | A | A | ? | N | A | ? | N | N | N | N | N | A | A | ? | ? | U | N | **28%** |
| 30 | | D | N | N | N | N | N | N | N | N | N | N | ? | N | N | N | N | N | N | ? | ? | ? | A | N | N | ? | ? | N | N | N | N | N | A | A | ? | ? | U | N | 8% |
| 31 | RNA DNA Shield | F | O | O | A | A | A | A | N | N | N | N | ? | N | N | N | ? | ? | N | N | N | N | ? | N | N | N | N | N | N | N | ? | N | A | A | ? | ? | ? | N | 22% |
| 32 | | RF | A | A | A | A | N | N | N | N | N | N | ? | N | N | N | ? | ? | N | N | ? | ? | ? | ? | ? | ? | ? | ? | ? | ? | ? | ? | A | A | ? | ? | ? | N | 17% |
| 33 | | RT | A | A | A | A | N | N | N | N | N | N | ? | N | N | N | ? | ? | N | N | ? | ? | ? | ? | ? | ? | ? | N | N | N | N | N | A | A | ? | ? | ? | N | 17% |
| 34 | | D | N | N | N | N | N | N | N | N | N | N | ? | N | N | N | ? | ? | N | N | ? | ? | ? | ? | ? | ? | ? | N | N | N | N | N | A | A | ? | ? | ? | N | 6% |
| 35 | TRIzol | F | U | U | A | A | N | N | N | N | N | N | ? | N | N | N | ? | ? | N | N | N | N | ? | N | N | N | N | N | N | N | ? | N | A | A | ? | ? | ? | N | 11% |
| 36 | | RF | U | U | A | A | N | N | N | N | N | N | ? | N | N | N | ? | ? | N | N | ? | ? | ? | ? | ? | ? | ? | ? | ? | ? | ? | ? | A | A | ? | ? | ? | N | 11% |
| 37 | | RT | U | U | A | A | N | N | N | N | N | N | ? | N | N | N | ? | ? | N | N | ? | ? | ? | ? | ? | ? | ? | N | N | N | N | N | A | A | ? | ? | ? | N | 11% |
| 38 | | D | N | N | N | N | N | N | N | N | N | N | ? | N | N | N | ? | ? | N | N | ? | ? | ? | ? | ? | ? | ? | N | N | N | N | N | A | A | ? | ? | ? | N | 6% |
| 39 | Formalin | F | ? | ? | ? | ? | ? | ? | N | N | N | N | N | N | N | N | ? | N | N | N | N | N | A | N | N | N | N | U | N | N | U | N | A | A | ? | ? | ? | U | 8% |
| 40 | | RF | ? | ? | ? | ? | ? | ? | N | N | N | N | N | N | N | N | ? | N | N | N | A | A | A | ? | A | A | ? | U | A | A | A | A | A | A | ? | ? | ? | U | **31%** |
| 41 | | RT | ? | ? | ? | ? | ? | ? | N | N | N | N | N | N | N | N | ? | N | N | N | A | A | A | ? | N | A | ? | U | A | A | N | U | A | A | ? | ? | ? | U | 22% |
| 42 | | D | N | N | N | N | ? | ? | N | N | N | N | N | N | N | N | ? | N | N | N | ? | ? | A | N | N | ? | N | N | N | N | N | N | A | A | ? | ? | ? | N | 8% |
| 43 | Formaldehyde | F | ? | ? | ? | ? | ? | ? | N | N | N | N | N | N | N | N | ? | N | N | N | N | N | A | N | N | N | N | U | N | N | U | N | A | A | ? | ? | ? | U | 8% |
| 44 | | RF | ? | ? | ? | ? | ? | ? | N | N | N | N | N | N | N | N | ? | N | N | N | U | U | U | U | U | U | U | A | A | U | U | ? | A | A | ? | ? | ? | U | 11% |
| 45 | | RT | ? | ? | ? | ? | N | N | N | N | N | N | N | N | N | N | ? | N | N | N | U | U | U | U | U | U | U | A | U | U | N | N | A | A | ? | ? | ? | U | 8% |
| 46 | | D | N | N | N | N | N | N | N | N | N | N | N | N | N | N | ? | N | N | N | ? | ? | U | N | N | ? | N | N | N | N | N | N | A | A | ? | ? | ? | N | 6% |
| 47 | Paraformaldehyde | F | A | A | ? | ? | U | U | N | N | N | N | N | N | N | N | ? | N | N | ? | N | N | A | N | N | N | N | U | N | N | U | N | A | A | ? | ? | ? | U | 14% |
| 48 | | RF | A | A | ? | ? | N | N | N | N | N | N | N | N | N | N | ? | N | N | ? | A | A | A | A | A | A | O | O | A | U | O | A | A | A | ? | ? | ? | U | **42%** |
| 49 | | RT | A | A | ? | ? | N | N | N | N | N | N | N | N | N | N | ? | N | N | ? | U | U | A | U | U | U | U | U | U | U | N | U | A | A | ? | ? | ? | U | 14% |
| 50 | | D | N | N | N | N | N | N | N | N | N | N | N | N | N | N | ? | N | N | ? | ? | ? | A | N | ? | ? | N | N | N | N | N | N | A | A | ? | ? | ? | N | 8% |
| 51 | Gluteraldehyde | F | ? | ? | ? | ? | ? | ? | N | N | N | N | N | N | N | N | ? | N | N | ? | N | N | A | N | N | N | N | U | N | N | U | N | A | A | ? | ? | ? | U | 8% |
| 52 | | RF | ? | ? | ? | ? | ? | ? | N | N | N | N | N | N | N | N | ? | N | N | ? | O | O | A | O | A | A | O | O | A | U | O | A | A | A | ? | ? | ? | U | 36% |
| 53 | | RT | ? | ? | ? | ? | ? | ? | N | N | N | N | N | N | N | N | ? | N | N | ? | U | U | A | U | U | U | U | U | U | U | N | U | A | A | ? | ? | ? | U | 8% |
| 54 | | D | N | N | N | N | ? | ? | N | N | N | N | N | N | N | N | ? | N | N | ? | ? | ? | A | N | ? | ? | N | N | N | N | N | N | A | A | ? | ? | ? | N | 8% |
| 55 | Bleach | F | N | N | N | N | N | N | N | N | N | N | N | N | N | N | N | N | N | N | N | N | A | N | N | N | N | N | N | N | N | N | A | A | ? | N | A | N | 11% |
| 56 | | RF | N | N | N | N | N | N | N | N | N | N | N | N | N | N | N | N | N | N | N | N | A | N | N | N | N | N | N | N | N | N | A | A | ? | N | A | N | 11% |
| 57 | | RT | N | N | N | N | N | N | N | N | N | N | N | N | N | N | N | N | N | N | N | N | O | N | N | N | N | N | N | N | N | N | A | A | ? | N | O | N | 11% |
| 58 | | D | N | N | N | N | N | N | N | N | N | N | N | N | N | N | N | N | N | N | N | N | A | N | N | N | N | N | N | N | N | N | O | O | ? | N | U | N | 8% |
| 59 | *Percent Optimal or Acceptable →* | | **28%** | **28%** | 25% | 25% | 9% | 9% | 6% | 8% | 6% | 6% | 2% | 4% | 6% | 6% | 6% | **25%** | **25%** | 6% | 19% | 19% | 77% | 4% | 6% | 19% | 4% | 8% | 9% | 4% | 6% | 6% | **100%** | **100%** | 25% | 25% | 8% | 4% | |
| 60 | *Percent Unknowns →* | | 23% | 23% | 28% | 28% | 28% | 28% | 0% | 0% | 23% | 0% | **57%** | 6% | 0% | 6% | **51%** | 15% | 0% | 45% | 25% | 25% | 17% | 26% | 21% | 25% | 32% | 4% | 4% | 4% | 8% | 8% | 0% | 0% | **75%** | **68%** | **83%** | 0% | |

| KEY | |
|---|---|
| RT | Room Temperature |
| F | Frozen |
| RF | Refrigerated |
| D | Dried > 60 ℃ |
| CH | Coral host tissue |
| AS | Symbiodiniaceae & Other Algal Symbionts |
| M | Microbiome (bacteria, archaea, & viruses) - aseptic techniques and sterile tools required |
| SK | Skeleton |
| G | Gametes & early life stages |
| O | Optimal - The best possible or most favorable technique |
| A | Acceptable - a suitable technique for the given sample analyses |
| U | Undesirable - an acceptable technique, though sub-optimal |
| N | Not acceptable - not a suitable technique for the given sample analyses |
| ? | Unknown - there exists no known published supporting literature on a given technique |

13 0.722

**Figure 2 Methodological pipeline used during the preservation and archiving of coral specimens for research and restoration purposes for various downstream analyses.** The orange columns on the left-side of the figure (*i.e.*, columns B and C) indicate the methods used during the collection and archiving of coral fragments, categorized by the chemical preservatives and fixatives used and the method of temperature storage (*i.e.*, room temperature (RT), frozen (F), refrigerated (RF), or oven dried (D)). For example, a specimen collected using the row 7 pipeline is immediately frozen using liquid nitrogen, and then subsequently stored in a conventional freezer (*e.g.*, −80 °C) whereas a specimen collected using the row 20 pipeline is first stored in ethanol and then placed in a refrigerator at 4 °C. The remaining columns (*i.e.*, columns D–AM) describe whether a specimen collected using a given pipeline is suitable for a variety of downstream measurements such as DNA analyses or chlorophyll quantification. Downstream analyses are categorized into three disciplines: (1) Omics, (2) Physiology, and (3) Microscopy and Imaging (*i.e.*, row 2). These columns are further subdivided based on the specific type of coral material being used (*i.e.*, coral host tissue (CH), algal symbionts (AS), microbiome (M), skeleton (SK) or gametes (G)). Five levels of appropriateness are herein described: Optimal (O), Acceptable (A), Undesirable (U), not acceptable (N) and unknown (?). These designations are based upon publish methodological data as well as the consensus scientific opinions of 30 coral scientists who attended the Coral Bleaching Research Coordination Network meeting in June 2020. The percentage of downstream analyses which were afforded an optimal or acceptable appropriate designation is shown in column AN. The total number of potential pipelines that are acceptable or optimal for a given downstream analysis are shown in row 59. The total number of potential pipelines for which the suitability is unknown is shown in row 60.                                                

*Riginos et al., 2020*)). Going forward, implementation of specific collection, preservation, and archiving pipelines developed herein could further maximize and foster more collaboration among diverse community members and stakeholders.

## Consolidating methods for restoration specimens

Coral restoration and rehabilitation programs aim to assist in the recovery of reef ecosystems through passive and active means, and for the ultimate goal of creating a reef that can independently continue to develop without further intervention (*Boström-Einarsson et al., 2020*). Recent efforts to explore the success and failure of some restoration programs have revealed a lack of coordinated efforts among restoration practitioners, scientists, and managers. Further, some restoration programs remain unlinked to scientific endeavors that could track natural biological, chemical, and oceanographic phenomena that provide mechanistic context for why some coral propagation and outplanting efforts result in success while others do not. Collaborative work to engage in scientific inquiries before, during, and after restoration efforts, along with standardized practices, could accelerate and advance restoration programs.

For example, genetic, physiological, and microbiome sampling of specimens from restoration corals that are successfully outplanted have revealed key aspects of why some genotypes and species are more resistant or resilient to local and global stressors (*Baums, 2008*; *Lohr & Patterson, 2017*; *Morikawa & Palumbi, 2019*; *Klinges et al., 2020*; *van Woesik et al., 2021*; *Voolstra et al., 2021*). Thus, the consolidated methods presented herein can be used to bridge the gaps between the restoration and research communities more readily and completely.

## General considerations for collecting, preserving, and archiving coral bleaching specimens

The central aim of our workshop was to identify simple and low-cost methods within the three broad categories of Omics, Physiology, and Microscopy and Imaging analyses that could increase the impact of every coral bleaching study in an effort to best understand scientific principles and increase restoration and conservation success. In the process, we

uncovered several key issues that all researchers and managers can consider regardless of individual subfields, including: (1) specimen and sample provenance and metadata, (2) sample collection considerations, and (3) sample handling and storage considerations. It is also important to consider how collection, preservation, and storage methods may shift the accuracy or precision of downstream analyses. For a more elaborate discussion of specific methods see the Supplemental Materials.

### Specimen/Sample provenance and metadata

Museums, research aquariums, and private collections have standard protocols for documenting the history, or origin, of individual specimen (*Smithsonian Institution, 2006*; *National Science & Technology Council, Interagency Working Group on Scientific Collections, 2009*; *National Academies of Sciences, Engineering, & Medicine, 2020*). Researchers and practitioners can optimize the use of their data and samples by rigorously cataloguing, and formally documenting as many experimental (*e.g.*, temperature ramp rate, light level, flow), biological (*e.g.*, coral color, morphotype, taxonomy, provenance), and environmental (*e.g.*, depth, nutrient concentrations, reef type) variables as possible (*Grottoli et al., 2021*) because these data provide needed context for each collection. We refer to these descriptive, contextual data as metadata.

Representative samples can also be properly 'vouchered' with a museum for long-term preservation, and such specimen can have important applications for a wide range of future work from these biological collections. First, if such samples include both tissue and the taxonomically informative skeleton, they can provide a taxonomic reference in the event that cryptic species are discovered, or to assign identity of the samples with future changes to taxonomy. Such vouchered samples also provide invaluable reference barcodes for databases that are becoming increasingly important as environmental DNA (eDNA) approaches become commonplace. Likewise, techniques change through time and questions that would have been impossible to address from such samples a couple of decades ago have become common place today with the advent of high-throughput and single-cell sequencing. Finally, even in cases where there is no obvious need to preserve the samples, the value of having historical samples has been showcased repeatedly in the field of epidemiology and emerging zoonotic disease research, where natural history collections have been integrated with host-pathogen research to resolve pathways of transmission (*Thompson et al., 2021*). The questions that will be answered by historical samples may yet be unknown, but it is only possible to address them if the samples are collected, vouchered and properly maintained.

Sample provenance also includes the documentation of how and where samples and their resulting data and metadata are physically and digitally stored. Growing recognition of the value of historical data and appreciation for FAIR (findability, accessibility, interoperability, and reusability) data standards (*Wilkinson et al., 2016*) is inspiring the efforts to archive sample data and metadata in ways that facilitate reuse and ensure archived data is available to future researchers (*Zerbino et al., 2018*; *Davis et al., 2019*; *Percie du Sert et al., 2020*). For example, the Genomic Observatories Metadatabase (*Deck et al., 2017*), stores metadata archives permanently linked to -omics resources stored at the

National Center of Biotechnology Information's (NCBI, Bethesda, MD, USA) and the National Science Foundation's Data Management Office (BCO-DMO) serve as repositories where samples and associated metadata are linked to the researchers who produced those studies and can be contacted about collaboration or specimen sharing.

Last, in regard to data provenance, many funding agencies have specific data management and dissemination requirements (*e.g.*, BCO-DMO at the National Science Foundation, GenBank at the National Center for Biotechnology Information, Environmental Data Service at the Natural Environment Research Council). However, relevant details concerning these samples and their province legacy data are often overlooked by researchers. For example, a recent sampling of the Sequence Read Archive (SRA) of GenBank found that only ~14% of all archived specimens associated with a sequencing project included both collection year and site as basic metadata that would be required for the reuse of archived genomic data in future studies (*Toczydlowski et al., 2021*). As the culture of global research and reef conservation and restoration have moved toward more open and collaborative models, there is growing pressure from funding bodies, journals, management agencies, and researchers alike to provide these data in open-access formats (*Sibbett, Rieseberg & Narum, 2020*), and develop community-wide cyberinfrastructure that facilitates the discovery and reuse of material samples (*e.g.*, iSamples; *Davies et al., 2021*). Such consolidated efforts stand to benefit the advancement and accessibility of the field of coral bleaching research and restoration science and effort as a whole.

### Specimen/Sample collection considerations

There is a myriad of possible techniques for collecting, processing, and archiving most coral specimens (for more details see Supplemental Materials). However, unique differences among coral taxa including individual colonies, their morphotypes, tissue thicknesses, skeletal density, and variation in life history demand special consideration as these variables may affect the biology and chemistry of collected coral samples and could dictate the applicability of many downstream procedures. Additionally, colony and specimen/sample size as well as species-specific variation can affect how corals respond to and recover from stress (*Brandt, 2009*; *Thomas & Palumbi, 2017*; *Álvarez-Noriega et al., 2018*; *Levas et al., 2018*). The quantity of available sample material can also affect what downstream techniques are possible. Improving the precision of measurements of colony and specimen size is an active area of research (Table S1) with the advent of new technological developments such as 3D laser scanning and photogrammetry (*House et al., 2018*; *Vivian et al., 2019*; *Zawada, Dornelas & Madin, 2019*). Information about the original size of the parent colony or outplant specimen can provide helpful information for interpreting resulting data because size has been shown to be an important bleaching predictor (*Álvarez-Noriega et al., 2018*).

Collection permits may also restrict the number of samples that can be collected, which can affect the types of analytical methods that are possible downstream and how much excess material may or may not be available for archiving and future research. Further, some agencies restrict the use of specimen for explicitly defined goals within a given project

or on a specific permit and thus may be unavailable for alternative end goals. Researchers should be aware of such agency-based limitations and enquire with providers on any downstream use restrictions. Last, developmental stage can have significant impacts on which methods are suitable and practical for any methodological pipeline. For example, the amount of material required for some analyses may be prohibitive when working with coral larvae or gametes, but easily performed on adult tissues. Thus, the types of research questions that can be addressed will vary depending on the life stage of the specimen and dictate the types of downstream analyses and collaborations that are most productive.

### Temperature and sample storage considerations

When collecting and preserving coral bleaching and restoration specimens for short and long-term use, documenting a sample's temperature history is critical (see Box 1 on freezing and cryopreservation). In general, altered temperatures can cause rapid state changes in live specimen physiology, microbiology, and geochemistry. Many subcellular and cellular processes can change within minutes to hours when corals undergo shifts in ambient temperature (*Hillyer et al., 2017a*), and swift sample processing is important to capture those responses. Once samples are preserved, temperature can further influence the integrity of each sample for some types of analyses. For example, cells lyse if samples are too cold, thus making them unsuitable for imaging of intact cells. Each scientific discipline (Omics, Physiology, and Microscopy and Imaging) has guidelines for optimal preservation temperatures suitable to ensure the integrity for their analytic process (see Fig. 2). The duration of storage for these specimens can also dictate ideal archiving temperature conditions. If samples are intended to be stored for tens of years (*e.g.*, in coral gamete biobanks), cryopreservation and downstream restoration, rapid-freezing in liquid nitrogen, and storing at −80 °C are the safest holding temperatures. If the tissues or cells can tolerate freeze-drying, and the final packaging is vacuum-sealed, then such specimens can be maintained for many years at room temperature.

However, coral bleaching and restoration research is often conducted in locations where adequate freezing agents and materials (*e.g.*, liquid nitrogen, dry shippers, or even ice) may not always be available or reliable. Although not all methods require temperature stabilization, many do (Fig. 2). Therefore, if possible, all researchers should record above and below water (1) the transport holding temperature, (2) any altered temperatures during transport, and (3) the duration of transport. For example, if live or dead specimens were removed from an offshore reef, transported to shore, and placed in new containment, the method and duration of transport as well as the temperature of any onshore activities (*e.g.*, freezer storage, water temperature manipulation) should be documented and included in the sample metadata.

### Specimen handling and sterility considerations

There is increased interest in how the coral holobiont microbiome (*i.e.*, Symbiodiniaceae, bacteria, viruses, and other microscopic eukaryotes) responds to, and may be involved in, preventing or exacerbating coral bleaching and/or increasing or reducing restoration success. Many ecological and physiological bleaching studies can be easily paired with

> **Box 1  Freeze it and forget it?**
>
> Freezing material is at the heart of maintaining robust tissue archives. But what are the consequences of some of these freezing processes in terms of tissue quality over time? Before deciding how to store samples, both the sensitivity of the downstream analysis (*e.g.*, RNA stability) and how long that process needs to be viable should be considered. The cryopreservation field is rapidly evolving, especially for human samples. For example, standard practice for understanding tumor physiology was to fix in formalin, embed in paraffin, and store at room temperature. However, delicate RNA can degrade over time under these conditions but remains robust if stored at −80 °C (*Baena-Del Valle et al., 2017*). Thus, coral RNA and enzyme specimens may best be stored at −80 °C, potentially remaining stable for up to 10 years at these low temperatures and making them suitable for additional downstream analyses. For corals, storing at −80 °C allows for the highest number of downstream analyses (Fig. 2). However, longer-term stable storage (>tens of years) at liquid nitrogen temperatures (−196 °C) is preferable (*Ortega-Pinazo et al., 2019*; *Kelly et al., 2019*), though highly impractical for many researchers due to the cost and equipment needs associated with ultra-cold storage. In contrast, many laboratory analyses can be reliably performed on specimens stored at −20 °C (Fig. 2) for 2 to 5 years.
>
> *Frozen But Alive: Cryopreservation Holds Material Safely for Many Years*
>
> Cryobiology is the study of cells and tissues at cold temperatures. The central principle in cryopreservation is to avoid the formation of lethal intracellular ice. Generally, cryopreservation uses permeating cryoprotectants or solutes, such as dimethyl sulfoxide (DMSO), methanol or propylene glycol, and non-permeating solutes, such as sugars (*e.g.*, glycerol), to allow the permeating cryoprotectants to enter cells and block ice crystal formation, and to permit the non-permeating solutes to dehydrate and remove intracellular water to reduce and avoid ice formation. Once cells and tissues are safely cryopreserved and held at liquid nitrogen temperatures, most biological processes are reduced. Theoretically, if cells are maintained at liquid nitrogen temperatures, they can survive for thousands of years with minimal damage. Thus, cryopreservation of living coral tissue and maintenance in liquid nitrogen (*e.g.*, cryobanks) provides access to a multitude of scientific and restoration uses because the tissues are frozen, but also alive. Once the cryoprotectants are warmed and the cells are rehydrated, they are alive, and any number of analyses can be done post-thawing. However, cryopreserved cultured cells are equally robust at either −196 °C or −80 °C using a number of metrics over 8 years (*Miyamoto et al., 2018*). Even in properly cryopreserved samples, tissue degradation can occur if samples are removed from a freezer to subsample and then refrozen or exposed to heat transients by opening and closing of a freezer door. Thus, avoiding any changes in freezer temperatures is ideal.
>
> To date, cryopreservation processes have been used to preserve coral sperm from over 48 species worldwide (*Hagedorn et al., 2012*). This international collaboration has used frozen sperm to subsequently fertilize coral eggs and create new coral larvae (*Hagedorn et al., 2017*). Moreover, frozen sperm has also been used to demonstrate the feasibility of assisted gene flow in the critically threatened coral *Acropora palmata* (*Hagedorn et al., 2021*). Frozen coral material is now archived in biorepositories around the world, and some of the material for the assisted gene flow experiments was stored for up to 10 years before successful use in fertilization experiments.

Symbiodiniaceae analyses (*e.g.*, cell densities, gene sequencing) through shared samples, but the potential for coral bacterial and/or viral analyses is severely compromised when sterile collection tools (*e.g.*, gloves, bone-cutters) and sterile receptacles (bags or tubes) are not used. The use of aseptic handling techniques during coral collection and processing is a relatively small and inexpensive change in the methodological pipeline that can enable additional downstream microbiome analyses (Fig. 3). For example, a suitable aseptic technique in the field may be as simple as wearing nitrile gloves when handling corals and using sterile receptacles, such as Whirl-Pak sample bags, and minimizing cross-contamination by replacing with clean equipment or sterilizing in between handling different specimens. For microbiome work, additional sampling of the environment (*e.g.*, water and sediments) can provide information about sources of potential contamination.

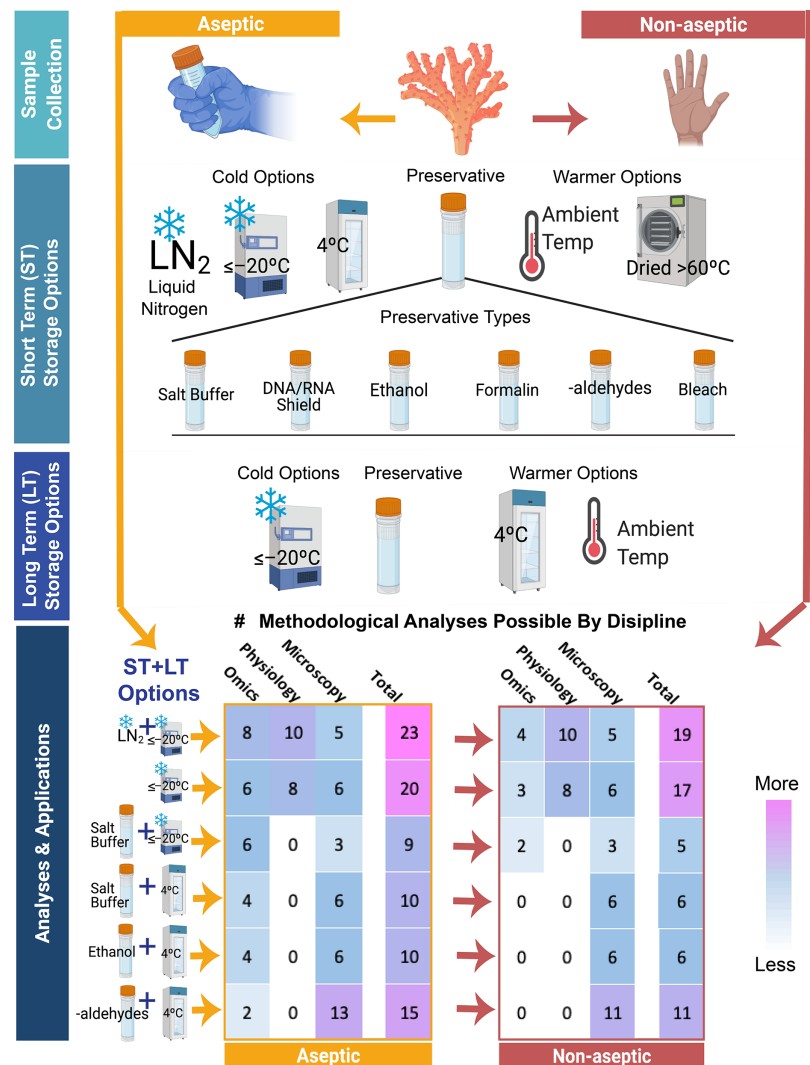

**Figure 3 Pictogram outlining and quantifying some of the most commonly used sampling and preservation pipelines for exploring questions about coral bleaching and restoration.** Shown in the decision tree are (1) the choice of different steps in coral specimen handling (aseptic on the left and non-aseptic on the right) (2) the range of initial short term storage techniques (ST) which include different types of temperature storage or chemical preservation and the range of preservation methods (3) and a selection of long-term (LT) storage options. Using summary data from Fig. 2, we calculated the total numbers of possible downstream analyses (values in colored boxes, ranging from fewer techniques (blue) to higher (bright pink)) that can be conducted for the three primary disciplines discussed in this review (omics, physiology, and microscopy techniques) when different combinations of aseptic or septic techniques along with different ST and LT options are chosen. The use of aseptic techniques (depicted on the left-hand side) in sample collection, preservation, storage, and archiving increases the number of possible downstream analyses shown at the bottom, relative to specimens handled using non-aseptic tools and receptacles (right-hand side), particularly in the Omics category. Similarly, freezing of samples at any point in the pipeline may limit the number of Microscopy and Imaging analyses that can be applied, though higher temperature storage points (>4 °C) and sample state changes limit the utility of specimens for many Omics and Physiological analyses as detailed in Fig. 2. Freezing of samples in liquid nitrogen (LN$_2$) or between –80 °C and –20 °C while using aseptic technique allows for the greatest number of the common downstream analyses depicted, represented by the bright pink cells. While many other applications and methods are possible, those depicted here are a small representation of common methods and not meant to be a fully inclusive list. Specific conditions for many analyses are more thoroughly outlined in the table of Fig. 2.                             

Importantly, while aseptic techniques are ideal for many downstream applications, it is impractical if not impossible to maintain underwater and in some handling situations.

### Caveats and considerations for methodologies, accuracy, and usability

In each discipline there may be recommended and, in some cases, well benchmarked standard operating protocols for each individual method discussed below and in the Supplemental. However, many of the methodological pipelines discussed below which are suitable for some aspects of coral-bleaching and restoration research but have not yet been fully evaluated in terms of their accuracy and precision. Therefore, deviations from standard procedures for a given discipline could potentially result in data that are inaccurate, uninterpretable, or unusable. It is important to consider the potential caveats when using any non-standard procedure in field and laboratory work. Yet, as research techniques improve and additional methods and protocols are confirmed as having high precision and accuracy, more of the potential pipelines discussed below may be employed with confidence in any given discipline. For example, using chemically fixed (*e.g.*, in formaldehyde) samples for genomic-based analysis has been considered non-standard in the past, but new work shows that these preserved specimens can be reliably used to gain insight into various aspects of coral biology, retrospectively (*Greene et al., 2020*).

### Identification of consolidated methodological pipelines for general use in coral bleaching and restoration studies

A previous literature review identified many methodologies in coral-bleaching studies (*McLachlan et al., 2021*), broadly categorized into three disciplinary areas: Omics (*e.g.*, genomics, epigenomics, transcriptomics, metagenomics, amplicon analysis, proteomics, and metabolomics), Physiology (*e.g.*, chlorophyll, lipid/protein/carbohydrate concentrations, biomass, tissue and skeletal stable isotopes), and Microscopy & Imaging-based analyses (*e.g.*, Symbiodiniaceae density measures, electron microscopy, histology, Raman spectroscopy). Using 36 defined analytical assessments (Fig. 2 columns), we quantitatively determine which methodological pipelines can maximize the number of downstream procedures across these three disciplinary areas. We assigned several broad categorical terms to determine whether a step in the pipeline was 'Optimal,' 'Acceptable,' 'Undesirable,' or 'Not Acceptable.' Steps within pipelines marked 'undesirable' indicate that there may be research to show the method is not ideal, or that it is illogical to pursue a particular pipeline based on past evidence. Thus, caution should be taken when evaluating these incomplete pipelines. Further, in many cases it was unclear if limitations existed for a particular downstream method or pipeline due to a lack of existing references, and thus we also designated many cells in the matrix as 'unknown' (Fig. 2). These 'unknowns' are likely to have resulted from insufficient testing or knowledge in a particular area as opposed to the method being truly unacceptable; testing these approaches may present fruitful areas for future research. We then summed the number of 'optimal' and 'acceptable' cells to determine which pipelines best served a given set of downstream methodologies.

In evaluating the various methodological approaches used in specimen collection, preservation, and archiving, we were able to identify several pipelines that maximize the

number of downstream analyses that are possible (Fig. 2 green cells; Fig. 3 fuchsia pink cells).

### Freezing or fixation methods dictate most methodological pipelines

Instantaneous freezing or 'rapid freezing' in liquid nitrogen upon initial collection followed by ultra-cold storage (*e.g.*, −80 °C) maximizes the number of possible downstream analyses (supports ~64% of 36 methods; Fig. 2 blue and green cells row 7). Analyses that could concurrently or sequentially be conducted after specimen rapid-freezing and cold storage fell primarily within the Omics and Physiology disciplines, while rapid freezing is inappropriate for most tissue Microscopy and Imaging because it alters tissue integrity (see Box 1). Freezing post-collection using −80 °C and more conventional −40 °C or −20 °C freezers were also suitable for several procedures within the Omics- and Physiology-based methods (supports ~83% of 18 Omics- and Physiology analytics), except for some RNA-based analytics, which always require immediate rapid freezing or preservation (*e.g.*, in RNAlater, TRIzol).

Within the Microscopy and Imaging discipline, preserving in paraformaldehyde followed by refrigeration allowed for the greatest number of downstream analytics (70% of all Microscopy and Imaging techniques, row 48), including some Omics methods. However, few if any of the Physiological methods could be conducted on samples stored in these aldehydes.

### Methodological considerations are needed to determine the suitability of some collecting, preserving and archiving sample pipelines

A few analyses, including metabolomics, quantification and identification of mycosporine-like amino acids (MAAs), soluble lipid analysis, and histology techniques stood out as highly restrictive in their requirements for initial and secondary storage methods. Such methodological limitations could be due to stringency in storage requirements or, as suggested by the large number of unknowns in Fig. 2, due to insufficient testing of potential alternative methods. Thus, we summed the number of 'unknown cells' to determine which methods had the most uncertainty in terms of how samples could be collected, preserved, and archived. Numerous methods had many 'unknowns' (Fig. 2, row 60) limiting our ability to find suitable additional pipelines to access outside of their standard procedures. For example, biomass quantification and Raman spectroscopy each respectively had 51% and 83% unknowns for the different possible methodological pipelines we tracked.

Below we discuss considerations specific to each major discipline: Physiology, Omics, and Microscopy and Imaging, given these methodological differences. Furthermore, we add more details about standard operating procedures for each of the major downstream analyses within Fig. 2 and throughout the Supplemental Material. While not an exhaustive list, we aimed to give researchers enough information to consider how to collect, preserve, and archive their specimens for many potential applications. We also recognize that methods are continuously evolving with the advent of new technologies. It is likely that newer, better methods will eventually become available and, thus, future researchers

should take steps to confirm that additional procedures have not become available following the publication of this work.

## CONSIDERATIONS FOR INDIVIDUAL FIELDS OF STUDY

### Omics methods

'Omics' are a collection of methods that focus on the identification, characterization, and quantification of macromolecules (*e.g.*, nucleic acids, proteins, lipids, and carbohydrates) and biochemical compounds (*e.g.*, metabolites and vitamins). Typically, high throughput procedures are applied, such as DNA and RNA sequencing, mass spectroscopy, liquid chromatography, mass and nuclear magnetic resonance spectroscopy, or X-ray crystallography. Such methods span approaches in proteomics, metabolomics, transcriptomics, meta-barcoding, phylogenetics, epigenetics, microbiology, and virology and often require computationally intensive bioinformatic analysis of large datasets.

#### *Sample collection*

In general, the use of aseptic techniques is strongly encouraged across all coral bleaching and restoration research collections, allowing for the most complete set of downstream analyses, including the characterization of the microbiome (which includes the coral virome) for which aseptic technique is necessary to ensure further analytical accuracy and integrity (Fig. 2). However, Omics methods targeting the coral host and/or Symbiodiniaceae do not require sterile tools and receptacles unless total Symbiodiniaceae community diversity is being examined in high resolution. Additionally, reagents and materials can contaminate samples with off-target cellular materials, foreign nucleic acids, viral particles, and exogenous chemicals. Moreover, compounds or enzymes that degrade, damage, or alter macromolecules and biological compounds (*e.g.*, proteases, R/DNAses) can disrupt or inhibit many molecular processes (*e.g.*, cations and polymerase chain reaction) needed to create Omic datasets or intermediate sample products (*e.g.*, sequencing libraries). Thus, care should be taken to avoid the use of non-sterile materials and/or reagents that are not certified as molecular biology grade.

#### *Preservation for short term storage*

Rapid freezing and storage at −80 °C is the optimal collection and storage technique for most Omics work such as genomics, metagenomics, proteomics, and RNA-Seq methods (see below). While RNA-based methods (*e.g.*, transcriptomics) are notoriously sensitive to initial collection and storage conditions and require rapid freezing or immediate storage in salt buffers, DNA is more stable and thus can be collected and preserved in a variety of conditions. Several methods (*e.g.*, DNA- and RNA-based host, symbiont, and microbiome approaches) can also be used when corals are initially preserved in salt buffers, some aldehydes, or DMSO (*Gaither et al., 2011*; *Gray, Pratte & Kellogg, 2013*; *Hernandez-Agreda, Leggat & Ainsworth, 2018*). These buffered specimens can be stored short-term at a variety of temperatures because the compounds stabilize the nucleic acids (*Hopwood, 1975*; *Seutin, White & Boag, 1991*; *Dawson, Raskoff & Jacobs, 1998*; *Douglas & Rogers, 1998*; *McKenzie, 2019*). For metabolomics, rapid freezing in liquid nitrogen (which inhibits metabolic processes) and storage at −80 °C is optimal and generally considered

best practice, especially if subsequent separation of host and symbiont is desired (*Lohr et al., 2019*) as storing samples in methanol immediately restricts analyses to holobiont metabolome due to the potential for salt contamination (*Hillyer et al., 2017b*). Rapid freezing and −80 °C storage is optimal for samples targeting protein analysis or comprehensive proteomics, although samples can also be rapidly processed (*via* crushing in liquid nitrogen or airbrushing/water-picking) and preserved in a protein extraction buffer containing protease inhibitors for short- and long-term storage (*e.g.*, *Barshis et al., 2010*; *Tisthammer et al., 2021*). However, for epigenetics, specific sub-applications have different sample preservation requirements (see Fig. 2). All in all, the approaches available for preserving specimens for Omics work are limited by the requirements of the most stringent aspect of the molecules under study (see Supplemental Material for details).

### Processing

The amount of material necessary for each downstream Omics procedure varies significantly and according to the target input/quality requirements of each analysis (*e.g.*, ~1 mg of extracted protein for proteomics). Procedures that use amplification steps (*e.g.*, metabarcoding of the bacterial and archaeal 16S rRNA, Symbiodiniaceae ITS2, or eukaryotic 18S) will require little material (~1 cm$^2$). Genomics, metagenomics, transcriptomics, epigenetics, proteomics, and metabolomics will require more starting material (*e.g.*, >1 cm$^2$) with typically higher quality standards. Single coral fragments are frequently analyzed with multiple different, complementary procedures (*e.g.*, genomics, transcriptomics, and metabarcoding).

### Archiving for long term storage

Almost all macromolecules in specimens are more stable long-term when ultra-frozen and many of them can be stored at freezing temperatures indefinitely if in the appropriate fixative or buffer. However, some methods such as Assay for Transposase-Accessible Chromatin using sequencing (ATAC-Seq)-based epigenetics require unusual storage conditions, such as cryopreservation in liquid nitrogen.

### Caveats

Delays or alterations in sampling, processing, and preserving any specimen for these Omics techniques may alter the accuracy and precision of the resulting analysis. However, many alternative preservation and storage methods (*e.g.*, formaldehyde) for Omics work remain untested or are incompletely benchmarked, thus it is unknown how these methods may affect downstream analyses. Future Omics research could scrutinize how these under-studied pipelines may alter a specimen's true biological and chemical composition in the short-term and when stored for long periods of time.

## Physiology methods

Coral physiological measurements (*e.g.*, changes in photosynthetic capacity, lipid profiles, and endosymbiont chlorophyll concentrations) are a staple of coral bleaching research, with 51% of coral bleaching studies published since 1992 measuring at least one physiological trait (*McLachlan et al., 2020*). Fig. 2 lists the 10 physiological measurements

most commonly performed. Rapid freezing in liquid nitrogen followed by storage in an ultra-cold freezer was the optimal practice across all physiological methods that we evaluated, although direct freezing at −40 °C to −80 °C is also acceptable. In all cases, common practices in collection and archiving exist for many measurements.

### Sample collection

In contrast to several of the microbial analyses described above (see Omics section), sample sterility and aseptic conditions are not an absolute requirement for downstream physiological assays. As a general guide, a coral ramet height >1 cm or tissue area >1 cm$^2$ are minimal requirements. However, larger fragment sizes (*i.e.*, several cm$^2$) are generally desirable to minimize edge effects associated with tissue heterogeneity and avoiding tissue damaged in the sampling process. Larger fragments also allow for multiple laboratory analyses and facilitate better cross-comparisons among corals and studies because stress conditions may decrease the amount of a given variable (*e.g.*, lipids), making it harder to measure in very small specimens. In addition, a small sample size (1 cm$^2$) may provide enough material for one analysis (*e.g.*, endosymbiont chlorophyll *a*), whereas considerably more material is required for other analyses (*e.g.*, coral tissue-based isotopes that can require four times this amount). When combining these requirements with the added benefit of long-term archiving for later potential use, investigators will often want to double their number of ramets when possible. Large fragments or ramets also have the benefit of providing a more representative sample and minimizing fragment edge effects associated with sampling, as well as any positional effects within the coral fragment itself (*e.g.*, top vs side of a branch or coral-mound). If the sample is not immediately processed at the time of collection, immediate rapid freezing in liquid nitrogen or immediate freezing at −80 °C followed by storage in liquid nitrogen or at −80 °C are ideal. Unless the goal is cryopreservation, in many cases freezing at −20 °C or colder is suitable for several physiological analyses.

### Preservation for short term storage (days-months)

Storage at −40 °C or colder for days to months is typically suitable for all physiological analyses, though some methods have additional requirements (Fig. 2). Storage at −20 °C for up to several months is also acceptable for many, but not all, analyses. In general, storage in liquid preservatives or fixatives (*e.g.*, methanol, formalin, *etc.*) is considered either not acceptable or the efficacy of such preservatives is unknown when considered for many types of physiological analyses. Freeze-drying was noted as a suitable method for storing samples for physiological assays and is beginning to be used in coral bleaching studies (*Wall et al., 2021*; *Pupier et al., 2021*; *Baumann et al., 2021*). Freeze-drying is especially conducive to isotopic analysis (*Wall et al., 2020*) and provides for easy storage and transport as freeze-dried samples can be stored at room temperature. Notably, these samples should be stored in the dark.

### Processing

For some physiological analyses, the processed subsample or specimen will be completely consumed and thus not available for long-term archiving. For example, a coral subsample

which is ground and burned in a muffle furnace for ash-free dry weight biomass quantification cannot subsequently be used for chlorophyll concentration analysis. Notable exceptions include lipid extracts, skeletal material prepared for elemental analyses, and cryopreserved samples, all of which can be archived long-term for additional downstream analyses. Nevertheless, given the desire for technical replication and repeatability, many investigators typically collect coral fragments large enough to have surplus materials that remains unprocessed and may be placed into a long-term archive for potential use in other downstream analyses (summarized in Fig. 3). A strategically designed workflow could therefore use one suitably sized fragment for multiple methods.

### Archiving for long term storage

Future physiological analyses are possible with frozen or freeze-dried fragments, ground whole coral, and frozen tissue homogenates and isolated fractions (*e.g.*, host and microbial fractions), dried skeletal material, and cryopreserved tissues. Except for analyses that rely on live samples (*e.g.*, cryobiology of reproductive cells and nanofragments; see section above on cryopreservation), all non-skeletal physiological analyses are possible with material that has been stored long-term at $-80\,°C$ or freeze-dried and then stored between $-80\,°C$ and $-40\,°C$. However, the efficacy and accuracy of using material that has been in long-term storage between $-20\,°C$ to $-40\,°C$ is currently unknown. Due to protein degradation, denaturation, or other decomposition, physiological analyses are not possible on coral material that has been stored long-term at $4\,°C$ or warmer. In most cases, long-term storage of more than a year is reasonable at temperatures of $-80\,°C$ or less, but it is unknown if some analyses could be reliably performed on material that was archived for more than 10 years (see above for cryopreservation where the efficacy of long-term storage is established, *e.g.*, sperm storage >10 years). In this regard, more study is required to determine the maximum duration that samples can be archived for each storage method (*i.e.*, liquid nitrogen, $-80\,°C$, and freeze-drying) and still be suitable for physiological analyses. An exception to this rule is with the coral skeleton (usually in the form of cores, cross-sections, ramets, or ground powder), which is best stored dry at room temperature or refrigerated at $4\,°C$ indefinitely so long as no future analysis of the skeletal organic matrix is intended.

### Caveats

For all but the skeletal isotopic and elemental analyses, the subsamples for each individual physiological analysis are drawn from a representative, homogeneous mixture of either ground coral or tissue blastate that is collected from a larger fragment in both height and surface area than what is needed to make the specific measurement. In addition, obtaining a larger sized fragment or subsample than what is strictly needed to conduct the analysis is recommended to obtain a representative sample, to minimize edge effects potentially associated with the way the fragment was cut, and any positional effects within the coral fragment itself (*e.g.*, top *vs* side of branch or mini mound).

## Microscopy and imaging methods

Microscopy and Imaging of coral specimens are essential to many aspects of coral-bleaching and restoration surveillance and experimentation, and 58% of bleaching studies published over the last 30 years utilized at least one Microscopy and Imaging technique (*McLachlan et al., 2020*). Imaging can be at the gross-colony or micro-corallite morphological level (Fig. 2), at the tissue or cellular level (often referred to as histology), or at the subcellular level using techniques like electron microscopy and Raman spectroscopy.

### Sample collection

Typically, coral samples collected from the field for Microscopy and Imaging should not be rapidly frozen, but instead placed in temporary storage, preferably in a cooler (with some fresh sea water), and out of direct sunlight. Microscopy and Imaging techniques differ from other methods in the requirement that samples must be as intact as possible (*e.g.*, cells not lysed, corallites undamaged, or lacking any fragment alteration). A variety of imaging techniques such as 3-D photogrammetry and CT-scanning for surface area must be conducted on intact colonies or fragments prior to any additional analyses on ground skeleton or blastates. Similarly, photographs for coral color (*e.g.*, to assess bleaching by loss of pigment) ideally are collected while the animal is alive before any other alteration to the colony has taken place and is thus best conducted underwater alongside a white standard or color chart. The need for intact skeletons and tissue extends to a variety of other Microscopy and Imaging techniques at smaller scales. Histology, XRF-scanning, Raman 2-D mapping, Nanoscale secondary ion mass spectrometry (NanoSIMS), and scanning electron microscopy (SEM) all require tissue or skeleton in their original shape or arrangement, on the scale of the analysis. For instance, using SEM to describe corallite morphology is possible on broken skeletal fragments, as long as those fragments are intact at the appropriate scale (*e.g.*, an entire corallite, which may be <1 mm to tens of mm depending on the coral species). There are, however, a few methods that can use ground skeleton or tissue (*e.g.*, Raman spot measurements, geochemistry), which can be conducted on altered or broken samples as long as they are suitably preserved. Investigations concerning the localization of nucleic acids, proteins, or microbes require immediate fixation upon collection.

### Preservation for short term storage

The short-term preservation of samples for Microscopy and Imaging varies with the objectives of the research. For analyses requiring tissue fixation, the type of fixative used varies greatly between researchers and applications, and there is no single best method. There are several commercial preservative kits available (*e.g.*, Bouin's, Z-fixed (buffered aqueous zinc formalin), Glutaraldehyde/Paraformaldehyde solution mix), and many preservative types that can be prepared (*e.g.*, formalin, glutaraldehyde, formaldehyde, methanol, ethanol, salt buffer). The proportion of each fixative (*e.g.*, paraformaldehyde and glutaraldehyde) and time of fixation can significantly affect the accessibility of epitopes for immunolocalization (for both light and electron microscopy) and may require extensive optimization for each target and tissue type. Over-fixation is of particular concern for small

samples like gametes and larvae. Techniques targeting nucleic acids (*e.g.*, fluorescence *in situ* hybridization (FISH)) also require special handling and electron microscopy (EM) grade or molecular grade (RNAse/DNAse-free) reagents (*Wada et al., 2016*). The temperature and time of preservation varies with the preservative used and the type of sample collected (*e.g.*, fragment, larvae, gamete).

By contrast, some preservation methods are simply unacceptable as they directly induce alterations that significantly affect the capability of researchers to adequately investigate the sample. Examples of this are histological and electron microscopy artefacts in coral tissue integrity and structure induced by freezing, changes in skeletal structure induced by chemical exposure, or the deterioration or alteration of nucleic acids when improperly cooled or stored in particular compounds or at incompatible temperatures.

For geochemistry-based methods, it is highly likely that any chemicals that are added to coral skeletons will change their chemical nature, therefore coral skeletons are best stored dry at room temperature or refrigerated at 4 °C when conducting geochemical focused imaging.

For imaging or light-based methods of Symbiodiniaceae quantification (*e.g.*, hemocytometry, coulter counter, flow cytometry), short term sample preservation (*e.g.*, of a tissue blastate) is recommended at 4 °C without fixation for two reasons: (1) freeze/thaw cycles can lyse symbiont cells, and (2) fixation can alter cell counts. However, coral fragments are commonly stored frozen (−20 °C to −80 °C) prior to tissue homogenization. Tissue homogenates can be stored frozen prior to analysis over the short term with the caveat that cell counts may be affected, and multiple freeze-thaw cycles are best avoided. Additionally, although cleaning skeletons (*e.g.*, with hydrogen peroxide or sodium hypochlorite bleach) have potential to alter the ratios of certain isotopes (*Grottoli et al., 2005*; *Holcomb et al., 2015*), there is sparse information available on the effects of preservatives on coral skeletons. However, we suggest that adding any form of chemical preservative prior to geochemical analyses should be done with caution because these chemicals could alter the composition of coral skeletons by adding contaminants or causing partial dissolution.

### Archiving for long term storage

Derivatives samples following application Microscopy and Imaging methods of coral skeletons are usually solid and exist in the form of cores, thin sections, or powder, and are best stored dry at room temperature or refrigerated at 4 °C. These can be kept indefinitely, although the true shelf life of each of these has not been thoroughly benchmarked for all downstream analyses. Coral tissues for histology-based methods are typically stored as fixed tissues embedded in paraffin blocks or sections mounted on microscopy slides and can be stored at room temperature or 4 °C. For immunolocalization and fluorescence *in situ* hybridization (FISH), these blocks can retain their quality indefinitely, whereas any generated thin sections from these blocks can deteriorate much more quickly (*Wakai et al., 2014*; *Alamri, Nam & Blancato, 2017*). Thin sections treated with dyes should be stored in the dark, whereas skeletons and tissue blocks are typically not light sensitive.

Derivatives of coral skeletons, in particular skeleton cores and thin sections, can be used for various downstream analyses, including skeletal imagining (*e.g.*, using CT scanning, X-ray, and XRF) and skeleton geochemical analysis (*e.g.*, Raman spectroscopy, isotopic and elemental analysis). Tissue-containing derivatives can be used for similar downstream analysis. There are no derivatives from the symbiont count techniques (*i.e.*, hemocytometry, flow cytometry, automated cell counter), as the small sub-sample volume is typically consumed by these methods, though any remaining original sample may be available for additional analyses depending on its storage method (*i.e.*, tissue blastate, Symbiodiniaceae pellet). Digital imagery produced from many of these techniques (*e.g.*, electron microscopy, CT scanning, X-ray, coral color analysis), may be used for different image analysis techniques downstream.

While most fixed tissue derivatives can be stored at 4 °C or room temperature for long periods of time, stained (*i.e.*, dyed) samples should be stored in the dark, and there is little known about the long-term preservation of samples for imaging. Long-term storage of coral tissues or homogenates for symbiont quantification that have not been chemically fixed is possible at −20 °C to −80 °C for >1 year if subsequential cell counts are performed with a hemocytometer, but not recommended at −20 °C due to Symbiodiniaceae cell degradation. Coral skeletons should be stored in a dry location at room temperature, and properly curated with metadata on collection dates, locations, water depths, *etc*. (*Reich et al., 2012*), and ideally with unique accession numbers.The preservation of photographic imagery, particularly those that document reef conditions during the previous century (*e.g.*, *Shinn & Kuffner, 2017*), is also an important community goal.

## Future considerations for the coral bleaching and restoration community members regarding collecting, preserving, and archiving of coral specimens

This work is intended to provide a consolidated resource regarding specimen collection, preservation, and storage for researchers and managers conducting current and future coral bleaching and restoration work. We identified methodological pipeline overlaps that can be leveraged to expand the utility of experiments and specimens, as well as provide opportunities for collaborations. Many of these consolidated workflows could also be used or adapted in coral related fields with similar end goals. Rapid freezing of samples in liquid nitrogen or placing at −80 °C to −20 °C is optimal for most Omics and Physiology studies with a few exceptions; however freezing samples removes the potential for many Microscopy and Imaging-based analyses due to the alteration of tissue integrity during freezing. The use of sterile gloves and receptacles during collection supports the downstream analysis of host-associated bacterial and viral communities. For Microscopy and Imaging, samples are best stored in aldehydes. Across all disciplines, the use of aseptic techniques during collection, preservation, and archiving maximizes the research potential of coral specimens and allows for the greatest number of possible downstream analyses.

We also found that many potential method amendments are either untested or have yet to be fully benchmarked. Thus, we recommend that researchers and funding agencies work together to explore additional methods. The 'unknowns' in our summary (Fig. 2) will

hopefully encourage the community at large to publish methodology reports that demonstrate both positive and negative results in their method development. Often only positive results are published, limiting our view of what has been attempted previously. At the same time, we recognize that there is no single method that can be used for all downstream analyses; specimens from single corals can be collected in a variety of ways and still maintain and enable future research possibilities. We recommend that, when possible, researchers and managers collect original samples in a way that will optimize as many downstream analyses as possible, such as other target methods mentioned here, regardless of the focus of each experiment. Clearly, this requires more planning, more materials, and the means to store and distribute samples. Archiving samples long-term requires more storage capacity, and thus would greatly benefit from development, design, generation, and maintenance of long-term storage banks and freezers that can house specimens for collaborations and future investigations using new methods that might provide greater insight into the causes, mechanisms, and consequences of coral bleaching as well as enhancing and potentially increasing the success and impact of restoration science. Any such archives/storehouses would also require an accompanying publicly available database of each specimen and their associated metadata so that researchers would be able to identify the most suitable samples for additional study.

## GLOSSARY

Airbrushing: the use of pressurized and focused air, sometimes accompanied with a liquid to remove the surface tissue of corals from the skeleton.

Archiving: temperature and/or chemical fixative or preservation techniques for samples post-processing for potential future use.

Aseptic techniques: laboratory practices, procedures, and methods used to keep equipment and samples free from contamination from living microorganisms and nucleic acids such as DNA or RNA.

Blastate: semi-liquid mixture of fine coral skeleton particles, tissue, and mucus, usually in combination with seawater or a chemical stabilizer/preservative.

Blue ice: regular ice (ice cubes, ice packs, *etc*.). Storage temperature at or near 0 °C.

Computed tomography (CT) scanning: a technique where skeleton is exposed to X-rays from multiple angles, and the resulting 2-dimensional X-ray images are processed to produce a 3-dimensional image of skeletal density.

Cryopreservation: a process where organelles, cells, tissues, extracellular matrix, organs, or any other biological constructs susceptible to damage caused by unregulated chemical kinetics are preserved by cooling to very low temperatures.

Destructive: causing irreparable damage, rendering any sample unusable for further analyses.

Downstream analysis: the eventual laboratory analysis of the variable(s) of interest (*e.g.*, chlorophyll concentration, lipid concentration, gene expression).

Epigenetics: the assessment of the modifications to the genome outside of the nucleic acid sequence. Often used to understand non-genetic mechanisms of acclimatization and/or plasticity.

Fixative: a chemical substance, such as formaldehyde or paraformaldehyde, used to preserve and/or stabilize some aspect (*e.g.*, protein and/or cellular structure) of a specimen.

Genet: a genetically unique coral colony or a collection of colonies (ramets) that can trace their ancestry back to the same sexual reproductive event (*i.e.*, they stem from the same settler and, hence, share the same genome) (*Baums et al., 2019*).

Genomics: analysis of partial or whole genomes including assessments of nucleotide sequence, genetic organization, and putative gene and gene family function.

Long-term storage: temperature and/or chemical fixative or preservation techniques for samples >24 h after initial collection.

Lyophilization (freeze-drying): a shelf stable method of preservation. Lyophilization or freeze-drying removes water from a sample while the sample is under vacuum. As such, ice can be changed directly from solid to vapor without passing through a liquid phase. After dehydration, samples are shelf stable and can be stored in the lab away from sunlight.

Metabolomics: the assessment of metabolites found in a specimen.

Metagenomics (or metatranscriptomics): methods aimed at analyzing partial or whole genomes from all organisms within a mixed community, including assessment of the composition and potential function of DNA (or RNA) found in a specimen. Often used to look at genetic potential, microbial community composition and function, and/or genetic background of a specimen.

Micro-XRF scanning: micro-X-ray fluorescence scanning is a non-destructive analysis for major and minor elements at down to 5 µm resolution by scanning the surface of skeletal slabs. XRF is based on the excitation of material with X-ray radiation and detection of the emitted fluorescence radiation spectrum whereby each element reacts at characteristic energy lines.

Mycosporin like amino-acids (MAA): metabolites induced by high-light and high wavelength light in diverse marine organisms in order to either absorb damaging UV rays (*e.g.*, act as sunscreens) or offset their effects (*e.g.*, act as antioxidants).

Nanoscale secondary ion mass spectrometry (Nano-SIMS): a method that uses primary ions beam to interact with sample surfaces resulting in the generation of secondary ions which can be analyzed for their specific mass.

Parent colony: coral colony growing on the reef from which specimens were removed.

Preservative: a chemical solution in which specimens are placed to avoid decay, such as ethanol >70% or a nucleic acid stabilizing salt buffer.

Proteomics: the high throughput analysis of peptides, proteins, and protein modifications from a given sample typically using either nuclear magnetic resonance or mass spectrometer techniques.

Provenance: a record of sample origin, collection, sampling, processing, and storage methods over lifetime usage of samples and their associated specimen.

Raman Spectroscopy: liquid, gas, or solid samples are exposed to a laser beam, and the changes in wavelength of the scattered light produces a Raman spectrum, which provides information about sample mineralogy and chemical composition.

Rapid Freezing ("flash" freezing): immediate sample preservation *via* freezing. In the field this may include placing samples on dry ice or liquid nitrogen. In the lab, this may include placing samples on dry ice, liquid nitrogen, or in an ultra-cold freezer.

Ramet: replicate fragments or colonies originating from the same genet.

Receptacle: a piece of laboratory equipment that receives and contains something (*e.g.*, test tube, vial, bottle). Synonyms: container, holder, vessel.

Sample (noun): (1) a representative part or single item from a larger whole or group (*e.g.*, fragment from a coral colony or a whole colony from a reef); (2) a finite part of a statistical population whose properties are studied to gain information about the whole.

Sample (verb): to take a sample of or from.

Scanning electron microscopy (SEM): high resolution microscopy using focused electron beans rasterized across a surface and visualization using the secondarily emitted electrons that result from the beam interacting with atoms on the surface.

Short-term storage: temperature and/or chemical fixative or preservation techniques for samples <24 h after initial collection.

Skeletal elemental analysis (non-isotopic): measurement of element ratios in coral skeletons, typically relative to Ca. Examples include Mg/Ca, Sr/Ca, and U/Ca.

Skeletal stable light isotopes: measurement of stable carbon, oxygen, and boron isotope ratios in coral skeleton.

Specimen processing: laboratory manipulation to prepare specimens for desired.

downstream analysis (*e.g.*, airbrushing, freeze-drying, tissue homogenizing).

Specimen: (1) an individual, item, or part considered typical of a group, class, or whole; (2) a portion or quantity of material for use in testing, examination, or study.

Specimen collection: the removal of coral specimens from the reef or from experimental tanks.

Specimen preservation: the method by which coral specimens are sacrificed, preserved, and stored immediately following collection (*e.g.*, rapid-freeze with liquid nitrogen and stored at −80 °C).

Sterile equipment: tools (*e.g.*, gloves, forceps, cotton swabs), receptacles (*e.g.*, test tubes, vials, bottles), and other equipment (*e.g.*, fume hood, laboratory work bench) which have been decontaminated, and thus, are free from living microorganisms and nucleic acids such as DNA or RNA.

Stable isotope analysis: measurement of stable carbon and nitrogen isotope ratios in a specimen.

Transcriptomics: the assessment of the composition and often function of mRNAs and sometimes small regulatory RNAs found in a specimen. Often used to look at physiological changes/responses to particular focal conditions and/or the phylogenetic placement of a specimen.

Transmission electron microscopy (TEM): high resolution microscopy using beams of electrons transmitted through a specimen and then captured for visualization on some device or material. For electron transmission samples are typically required to be ultra-thin sectioned at thicknesses of less than 100 nm.

Ultra-cold freezing: storage of samples at ultra-cold temperatures (−40 °C to −80 °C).

Water-piking: the use of a Water-Pik (oral irrigator device) to remove the tissue from a coral skeleton using a jet of high-pressure water.

XRF scanning: X-ray fluorescence scanning is a non-destructive analysis for major and minor elements at cm to mm down to 200 micron-scale resolution by scanning the surface of split skeletal cores or of skeletal slabs. XRF is based on the excitation of material with X-ray radiation and detection of the emitted fluorescence radiation spectrum whereby each element reacts at characteristic energy lines.

## ACKNOWLEDGEMENTS

Any use of trade, firm, or product names is for descriptive purposes only and does not imply endorsement by the U.S. Government.

### Funding

This workshop was funded by the National Science Foundation Division of Biological Oceanography #1838667 to Andrea Grottoli, Rob Toonan, Rob van Woeski, Mark Warner, and Rebecca Vega Thurber. The funders had no role in study design, data collection and analysis, decision to publish, or preparation of the manuscript.

### Grant Disclosures

The following grant information was disclosed by the authors:
National Science Foundation Division of Biological Oceanography: 1838667.

### Competing Interests

Robert J. Toonen is a PeerJ Section Editor. The authors declare that they have no competing interests.

### Author Contributions

- Rebecca Vega Thurber conceived and designed the experiments, performed the experiments, analyzed the data, prepared figures and/or tables, authored or reviewed drafts of the article, and approved the final draft.
- Emily R. Schmeltzer conceived and designed the experiments, performed the experiments, analyzed the data, prepared figures and/or tables, authored or reviewed drafts of the article, and approved the final draft.
- Andréa G. Grottoli conceived and designed the experiments, performed the experiments, analyzed the data, prepared figures and/or tables, authored or reviewed drafts of the article, and approved the final draft.
- Robert van Woesik conceived and designed the experiments, performed the experiments, authored or reviewed drafts of the article, and approved the final draft.
- Robert J. Toonen conceived and designed the experiments, performed the experiments, authored or reviewed drafts of the article, and approved the final draft.
- Mark Warner conceived and designed the experiments, performed the experiments, authored or reviewed drafts of the article, and approved the final draft.

- Kerri L. Dobson conceived and designed the experiments, performed the experiments, authored or reviewed drafts of the article, and approved the final draft.
- Rowan H. McLachlan conceived and designed the experiments, performed the experiments, analyzed the data, prepared figures and/or tables, authored or reviewed drafts of the article, and approved the final draft.
- Katie Barott performed the experiments, authored or reviewed drafts of the article, and approved the final draft.
- Daniel J. Barshis performed the experiments, authored or reviewed drafts of the article, and approved the final draft.
- Justin Baumann performed the experiments, authored or reviewed drafts of the article, and approved the final draft.
- Leila Chapron performed the experiments, authored or reviewed drafts of the article, and approved the final draft.
- David J. Combosch performed the experiments, authored or reviewed drafts of the article, and approved the final draft.
- Adrienne M. S. Correa performed the experiments, authored or reviewed drafts of the article, and approved the final draft.
- Thomas M. DeCarlo performed the experiments, authored or reviewed drafts of the article, and approved the final draft.
- Mary Hagedorn performed the experiments, authored or reviewed drafts of the article, and approved the final draft.
- Laetitia Hédouin performed the experiments, authored or reviewed drafts of the article, and approved the final draft.
- Kenneth Hoadley performed the experiments, authored or reviewed drafts of the article, and approved the final draft.
- Thomas Felis performed the experiments, authored or reviewed drafts of the article, and approved the final draft.
- Christine Ferrier-Pagès performed the experiments, authored or reviewed drafts of the article, and approved the final draft.
- Carly Kenkel performed the experiments, authored or reviewed drafts of the article, and approved the final draft.
- Ilsa B. Kuffner performed the experiments, authored or reviewed drafts of the article, and approved the final draft.
- Jennifer Matthews performed the experiments, authored or reviewed drafts of the article, and approved the final draft.
- Mónica Medina performed the experiments, authored or reviewed drafts of the article, and approved the final draft.
- Christopher Meyer performed the experiments, authored or reviewed drafts of the article, and approved the final draft.
- Corinna Oster performed the experiments, authored or reviewed drafts of the article, and approved the final draft.
- James Price performed the experiments, authored or reviewed drafts of the article, and approved the final draft.

- Hollie M. Putnam performed the experiments, authored or reviewed drafts of the article, and approved the final draft.
- Yvonne Sawall performed the experiments, authored or reviewed drafts of the article, and approved the final draft.

## Data Availability

This is a review and there is no raw data associated with it.

## Supplemental Information

Supplemental information for this article can be found online at http://dx.doi.org/10.7717/peerj.14176#supplemental-information.

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
