# Peer review of "Unified methods in collecting, preserving, and archiving coral bleaching and restoration specimens to increase sample utility and interdisciplinary collaboration"

_PeerJ, doi:10.7717/peerj.14176_

## Round 0.1 · original submission · Major Revisions

Dear Rebecca and co-authors,

I have now received three independent reviews of your study. My apologies for the extended time to get to this point, but it took some significant efforts to find reviewers who would accept to take the seemingly daunting task of reviewing this large body of work that includes most world experts as co-authors (77 reviewers had to be solicited...). Therefore, I'd like to commend and thank the three reviewers for their time and valuable comments. While they clearly recognized the quality and importance of this work, they have collectively raised a number of issues that will need to be addressed in your revised manuscript.

Overall, the reviewers have provided you with excellent suggestions on how to improve the manuscript, and I will be looking forward to receiving your revised version along with a point-by-point response to their comments.

With warm regards,
Xavier

·

Basic reporting

The paper "Unified methods in collecting, preserving, and archiving coral bleaching and restoration specimens to increase sample utility and interdisciplinary collaboration" is an useful contribution that goes beyond the type of study enunciated in its title.

I think that the manuscript is well written and structured. It has sufficient introduction and background and high quality graphics. However, I did have some trouble with Table 2.

First of all, it should be referred to as a figure, even though it is in text format. There is no caption in the text beyond the table title and I don’t believe it is as self-explanatory as the authors would like. I think more detail could be given in the caption so as to help readers navigate the table. Authors should also explain the difference between “U” and “?” for the latter is not referenced in the table’s legend. This legend may be suppressed and detailed in the figure’s caption, together with the explanation for the asterisk symbol.

In order to improve the figure’s quality, authors should suppress the spreadsheet column letters in the first row, care for the relative row height and vertical alignment of the characters within each cell and for the wrapping in the columns of STEP 2 row.

In the columns starting with “STEP”, they should simply replace the brief description by the step number for brevity. For instance, instead of writing: “STEP 2: Following collection &/or transport, specimen is placed in:”, it would be better to write “STEP 2: following STEP 1, specimen is placed in:” Color coding “STEP #” should probably help the reader recognize the 3 top rows as a sequence of events.

Asterisks should be moved close to the row title in the “Specimen type” column instead of being placed next to the letter ranking the adequacy of each technique (e.g. microbiome*) since any viable technique applied to certain specimen types will require sterile tools. I would also change the background color of optimal (“O”) techniques to dark blue so that it becomes more clearly distinguishable from acceptable techniques.

The column “Specimen type” and “Downstream analysis” should be swapped. The authors should left-align and capitalize the first letters, and suppress repeated names. For instance, “any holobiont member” should appear just once and horizontal lines, thinner and possibly colored in a lighter shade of gray, should be applied to set boundaries among different specimen types. This will make the table less busy.

The STEP 3 row was the most confusing. In the text, the authors make a distinction among the archiving adequacy of -20oC, -80oC and ultra-cold freezing, but they lumped everything into the “Frozen” column, instead of discriminating among them. They should break this column down into its three components and re-score those columns accordingly. This will add several new columns to the table, but they may be able to compress the whole table into the allotted space by suppressing the letters in the cells and narrowing all the columns. As stated above, changing “O” cells to blue may do away with the need to have letters in the cells.

Still on the STEP 3 row, I may be very mistaken, but it seems that workers dry samples using > 60oC ovens, but no one stores specimens in them. This is a step that is used in the preparation of coral skeletons. The same is valid for “Bleach”, which is not used for storage, but for specimen preparation. The authors should transfer skeleton preparation to another table, since the work flow with this particular type of tissue is very diverse from the other types of tissue included in the table. Finally, as a side note, in my experience, coral skeletons may become moldy even after being bleached and oven-dried for hours is stored in humid conditions, so the authors should probably make the distinction between climate control and ordinary storage. This may also be true for other archiving forms.

The same confusion between tissue processing using bleach and/or oven vs. preservation is present on Fig. 1. and it should be properly addressed. It looks like the authors attempted to split the figure in two halves, separating Aseptic vs. Non-aseptic procedures. This figure is also busy and I don’t if the arrows, the unlabeled screw-cap eppies or the snowflakes are of any help. My guess is that the authors would be better served if they represented the combination of preservation/storage/application as flow charts such as Aseptic -> LN2+20oC -> Omics. If arrows have thickness proportional to or are color-coded according to the number of (presumably) studies (once again, the figure caption does not make it clear) that employed this combination, it would become much clearer to the reader what is the most used pipeline in each type of study. Plus, the figure would become more compact if the resulting tree was represented in circular format. Many packages in R designed to deal with network data should easily produce such diagrams. These diagrams would actually be more traditional representation of pipelines than what they have assembled in figure/table 2.

Experimental design

No comment

Validity of the findings

If properly reformatted, this paper will actually become a handy contribution for the interested worker that may use it as a quick guide to design protocols or as a more robust reference to field and lab procedures. Because the authors suggest at the end of the paper that storing large amounts of samples may actually require sizable investments that should probably be best handled at a institutional level, and considering that the idea of this paper arose from a NSF sponsored workshop, I felt that that a more elaborate discussion of the possible roles of universities/museums was lacking, including ongoing initiatives that could allow non-affiliated researchers to collect more data from archived specimens. It looks like what the authors are proposing is the building of tissue banks akin to biological collections that have been used by generations of researchers interested in taxonomy, systematics, ecology, evolution, etc. On a minor note, I think that regardless of the application, researchers should also collect tissue (and maybe even skeleton) vouchers that could be used to confirm the taxonomic identity of the sample if needed (see https://journals.asm.org/doi/10.1128/mBio.02698-20 for a similar proposition). Perhaps the authors may find this another valid step in to include their pipelines.

Reviewer 2 ·

Basic reporting

Vega-Thurber et al. have produced a very useful primer on coral sample collection that will be invaluable to established reef scientists, and will likely save countless PhD chapters from ruin. The topic is important, the paper generally well-organised, and the content useful. It will be suitable publication a few minor issues are addressed, listed below.

The introduction section is too long, with too much space dedicated to justifying the work. The paper is the result of a workshop, and that is evident in the manuscript, which is very well-written but overlong. As the need has recently been discussed in a review by many of the same authors (McLachlan et al 2021), simply refer to the previous work. It would also benefit from more paragraph breaks throughout (especially the introduction).

The needs of permitting agencies and local stakeholders are only mentioned in passing and deserve a bit more discussion. Frequently, management agencies are only willing to allow collection of the minimum material needed to conduct the planned work, not for banking or for other potential uses that may arise later. In other cases local stakeholders, e.g. indigenous groups, may require consultation, approval, or have rights to samples collected (especially sequences/natural products). Changing or expanding a project may require that these groups be consulted again for their input or approval. Future users of banked samples or datasets will need to be aware of these requirements and any stored material have these needs attached and explained clearly. A brief mention of how planning for these scenarios can be incorporated into experimental plans and protocols would be useful. This could be added in a short paragraph after the “Specimen and Sample Provenance” section, line 205.

Line 466 – For metabolomics, rapid freezing with liquid nitrogen, followed by freeze drying, is also possible, and especially useful for sample transport (see Matthews et al. 2020, Metabolite pools of the reef building coral Montipora capitata…, Coral Reefs).

Line 478 – Mass spectrometry-based proteomics analysis typically requires ~1mg total protein as starting material (here or in supplemental). I also found discussion of proteomics in general to be underdeveloped in the manuscript.

573 – are essential

Supplemental material:
Line 316 – This is true, but gross overkill for some protein analysis methods. A simple Qubit fluorometer (or potentially a Nanodrop) can measure protein in the <10 microgram per mL range and only requires a few microliters.

Experimental design

N/A

Validity of the findings

N/A

Additional comments

N/A

·

Basic reporting

The manuscript on “Unified methods in collecting, preserving, and archiving coral bleaching and restoration specimens to increase sample utility and interdisciplinary collaboration” provides a comprehensive overview of methods applied to coral fragments/specimens by different disciplines, and how these samples could be collected and preserved to facilitate further usage and encourage scientific collaborations. The manuscript is written clearly and in professional English throughout, however, the way that the different sections are written varies substantially, which is not surprising for such a sort of review paper put together by several authors, but I think the text would profit a lot from some ‘leveling’ and ‘streamlining’. While some sections (particularly in the extensive supplementary materials) start with an introduction to the methods/approaches and explain what they are used for etc., others dive right into the practical procedures. Along with this goes a high variability in terms of background/context provided and literature referenced. Partly, references cannot be found in the literature list, and at some points, I had the impression authors were mostly citing their own work but it might be appropriate to cite more original studies developing the methods. The introduction to the idea and outline of the manuscript, describing the general intention and potential benefits from various viewpoints, is clear and more than sufficient.
As several of the methods start with the initial steps of collecting, (snap-)freezing, and airbrushing coral fragments, and are also often similar in how samples need to be stored, there is a lot of repetition. Hence, if possible, I suggest shortening overall, maybe merging some of the methods sections, and also parts of the main text, to make the text more reader-friendly and concise. I have highlighted paragraphs in the PDFs where I think there is an overlap with previous parts. To reduce repetition, I suggest considering combining collection & short-term storage, or short- and long-term storage.
Alternatively, instead of subdividing everything by disciplines and further by methods and parameters, which seems to inflate the text, could it be an option to structure the text by steps taken in a temporal order and highlight which ways the collect or store samples facilitate or disregard which further approaches? I could imagine a decision tree sort of flowchart that guides the reader / potential future coral scientist through the decisions on how to treat her/his samples, from minute one with systematically collecting and storing metadata, through the common collection, processing, and preservation options, to which results could be achieved from that. Likewise, it would be valuable to highlight this way which downstream analyses can still be conducted on partly processed samples, e.g. airbrushed tissue slurry, cleaned host fractions, or symbiont pellets.
Hence, from my point of view, the manuscript could profit from some restructuring, shortening, and making the different sections more consistent. The figures are nicely prepared but can be improved (see also comments in PDF), and Table 2 is unfortunately provided in such low resolution / small size, that I cannot judge its content.

Experimental design

I was happy to accept to review this manuscript, as I think it is a great and very useful idea. In recent experiments, I have been trying to come up with a workflow myself that would allow studying coral bleaching from a multitude of disciplinary perspectives, including ‘omics, physiological and geochemical methods, and hence I know that this can be quite a challenge. While doing so, I noticed the large variability even within the methods used to study one single parameter but was also positively surprised by how many parameters can in the end be measured on a single coral fragment if this has been thought through from the beginning. In the presented manuscript, I think an impressive amount of approaches has been summarized, but I am missing an overall synthesis/summary or suggested set of methods. While the aim of the article is to ‘unify methods’, and I am very aware that this is not an easy endeavor, I think there is still room for more ‘unification’. Some key points to consider which come up repeatedly could just be highlighted in a list of bullet points or a clear summary figure. All further comments are found in the attached PDF files.

Validity of the findings

no comment

---

## Round 0.2 · Minor Revisions

Dear Rebecca and co-authors,

Thank you for your thorough manuscript revision. Please find some final minor comments on the Figures made by Reviewer 1.

I'll be looking forward receiving your final version for acceptance. Please note that I'll be on annual leave without a computer from July 5th till August 2nd - I'll be able to handle the Revised manuscript either before or after these dates.

With warm regards,
Xavier

·

Basic reporting

I think that the authors did a good job overall addressing the multiple reviewer comments, although I saw no differences between the first and second versions of Fig. 3. I think it is poorly developed and may actually confuse rather than inform readers. For instance, the tables at the bottom of the figure have frequencies, but it is unclear if these numbers were tabulated from the published literature or simply by enumerating combinations of possible pipelines by some other means. It is also unclear if those frequencies are actually based on Fig. 2 or if they were compiled from other sources. Additionally, I still think that the figure is not as comprehensive as the authors intend it to be. For instance, Fig. 2 indicates that the optimal storage temperature for coral skeletons to be analyzed using CT scanning or X-ray imaging would be 60oC, but dry-ovens are not included among the long-term storage devices in Fig. 3. I personally think that this figure still need work, but I'll leave it up to the editor to decide what to do with it.

Experimental design

No comment

Validity of the findings

No comment

Additional comments

No comment

---

## Round 0.3 · accepted · Accept

Dear Rebecca and co-authors,

I am please to accept your manuscript for publication in PeerJ. Thank you for this great scientific contribution to the field.

With warm regards,
Xavier